# ONLY SPARSITY BASED LOSS FUNCTION FOR LEARNING REPRESENTATIONS

## ABSTRACT

We study the emergence of sparse representations in neural networks. We show that in unsupervised models with regularization, the emergence of sparsity is the result of the input data samples being distributed along highly non-linear or discontinuous manifold. We also derive a similar argument for discriminatively trained networks and present experiments to support this hypothesis. Based on our study of sparsity, we introduce a new loss function which can be used as regularization term for models like autoencoders and MLPs. Further, the same loss function can also be used as a cost function for an unsupervised single-layered neural network model for learning efficient representations.

## 1 INTRODUCTION

Sparse codes are an effective representation of data, that facilitates high representational capacity without compromising on fault tolerance and generalization. The advantages of sparse codes over both dense and entirely local coding schemes, has been discussed extensively in the computational neuroscience literature, Földiák & Young (1995); Tigreat (2017)), which contain ample evidence for the presence of sparse codes in cortical computations.

Sparse representations are also generated by many unsupervised learning methods. Recently, Konda et al. (2015b) showed that regularized autoencoders Rifai et al. (2011); Vincent et al. (2008) learn negative biases, and argued that this restricts the size of the regions in input space in which a hidden unit yields a non-zero response, effectively "tiling" the input space into small (but potentially overlapping) regions[1]. In this work, we present more general understanding on the relationship between the data distribution and sparsity of the learned representations. We hypothesize that when the data is distributed along a non-linear or discontinuous manifold, one of the ways to efficiently represent the data is by tiling the input space and let non-overlapping subsets of neurons represent the tiles which results in sparsity. We present a formal explanation for emergence of sparsity in hidden layers of networks trained on non-linear data distributions.

Based on our hypothesis for sparsity, we present a new regularization method for both unsupervised and supervised neural network models termed "One-Vs-Rest" (OVR). The regularization term computes overlap between hidden layer representations, in-terms of active dimensions, across input samples in a batch. Minimizing the OVR term as part of the cost function encourages sparsity in the hidden-layers, resulting in improved performance or more efficient representations, on-par with regularization techniques like dropout, denoising and contraction. In sparse autoencoders the objective of learning efficient representations for the input samples done by minimizing a regularization term along with reconstruction cost. We show that only the OVR term is enough as a cost function to learn sparse and efficient representations. This observation enables us to present a simple encoder (single layer) model using OVR-loss as cost function which can learn efficient sparse representations of the data without any decoding or reconstruction of input. The single-layer model presented is based on local Hebbian-like learning rules without any requirement of error back-propagation. In the following sections we present a more detailed explanation of our hypothesis, OVR term and the models introduced along with the experimental validation on some datasets.

## 2 SPARSITY IN NEURAL NETWORKS

Sparse representations can be observed in hidden layers of multi-layer-perceptron (MLP) as well as regularized autoencoder architectures. We in this section present a hypothesis for emergence of sparsity in hidden layers and show

---

[1]By this, we mean that the features learned by a neural network layer will cover small sub-regions of the input space, thus dividing it into tiles.

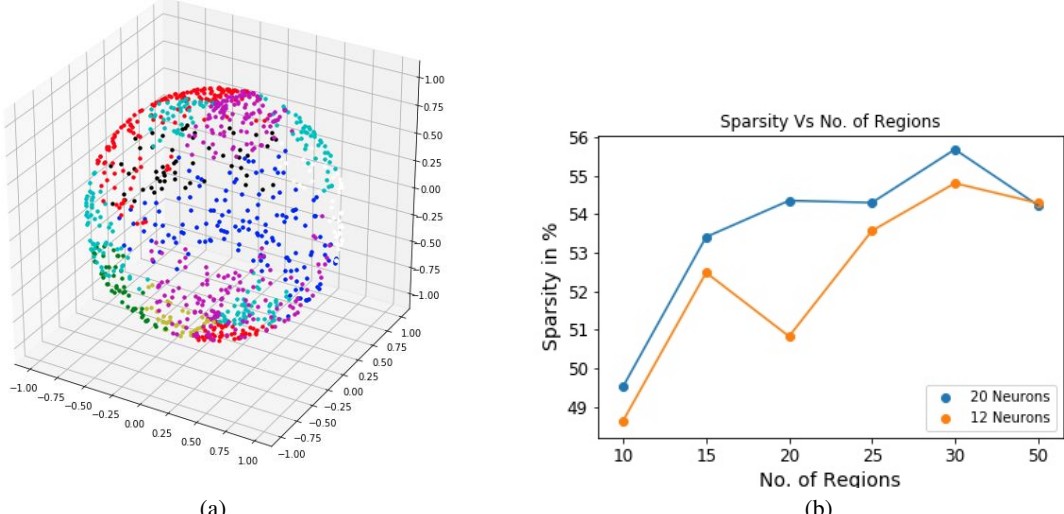

Figure 1: (a) Plot of the 3D points sampled from a distributions with 10 categories and 40 clusters (a discontinuous manifold). (b) Sparsity in hidden layer of classifier trained on these 3D data points. Legend in this plot shows hidden size of the network.

that it is related to non-linear or discontinuous nature of the input manifold and the number of neurons used in the hidden layer. We also present experiments to support our hypothesis.

## 2.1 WHY SPARSITY?

Let us assume that an input distribution is represented by $N$ different clusters, a discontinuous manifold. Given a set of $K$ feature vectors, an efficient representation of the clusters in feature vector space would be the one which maps all the input clusters to representations with least overlap (in terms of feature dimensions) across multiple clusters. Let $s_m$ be the set of active dimensions in the representation of a given cluster $M$. When we have a small $K$ and large $N$, which seems to be often the case in practice, the only way to generate enough representations to map all the clusters, such that the overlap across representations $s_i \cap s_j; i, j \in \{1, ..., N\}$ is small, is to reduce the average size of $s_m$, i.e., increase sparsity. When $K$ is insufficient to generate required number of representations with least overlap, it results in denser and inefficient representations with larger overlap.

To experimentally validate our hypothesis we created a toy dataset in which every sample is a 3D point on a unit-sphere. The sphere itself is divided into $N + 1$ latitudinal (horizontal) cross-sections and $M$ longitudinal (vertical) sectors. This arrangement divides the sphere's surface into $M * (N + 1)$ partitions. Each partition is randomly assigned one of the 'C' classes to make sure the data is distributed along a discontinuous manifold. For each experiment 5000 such points are generated belonging to 10 classes (C = 10). Single hidden layer neural network was trained on the dataset with varying number of hidden units. Sparsity of hidden representations from multiple experiments are presented in Figure 1(b). From the plot in Figure 1(b) it can be observed that the sparsity in the network increases with the increase in number of input clusters to be mapped.

## 3 A NEW REGULARIZATION TO ATTAIN SPARSITY

We propose a new regularization term named One-Vs-Rest (OVR) loss, based on the observation that sparsity can be achieved by forcing a neural network model to minimize the overlap between hidden representations of random samples from the dataset. Let $H$ be a matrix with the hidden layer representations of the samples from a randomly sampled mini-batch as rows. OVR-loss term is defined as,

$$\sum_{ij} HH^T. \tag{1}$$

Minimization of OVR-loss implies reducing the overlap across representations of samples, within the mini-batch, coming from different input clusters or regions. In the case where all samples in a mini-batch are from the same cluster

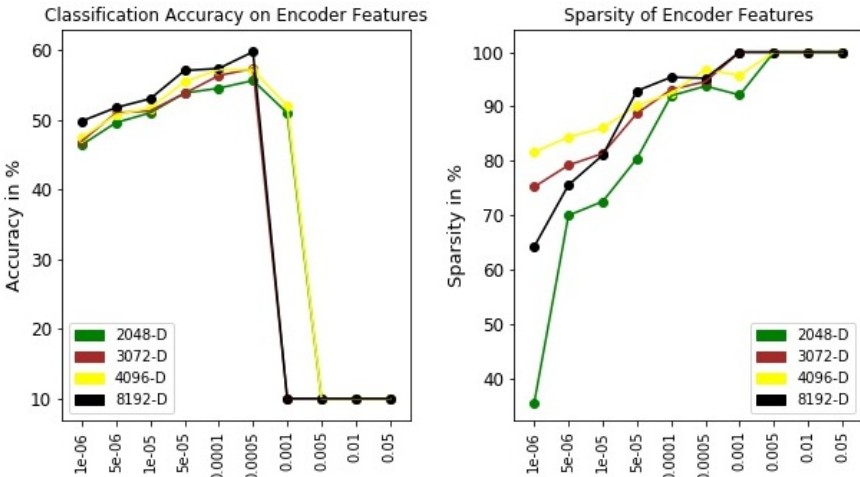

Figure 2: Sparsity of the representations and classification accuracy w.r.t regularization strength in autoencoder experiments, using ReLU activation and OVR-loss as regularization, trained on PCA reduced CIFAR-10. Legend in each plot shows number of hidden units in the experiments corresponding to the curves.

or region, highly unlikely when the batch is randomly sampled, OVR-loss cannot be used as it would mean learning different representations for samples from a local region of the input space, similar to the original data representation. OVR-loss can be employed as regularization term in both autoencoder and MLP architectures simply by adding the OVR-loss to the objective function of the model via a hyper-parameter $\lambda$.

## 3.1 AUTOENCODER AND MLP EXPERIMENTS

We ran multiple autoencoder experiments to evaluate OVR-loss for regularization. In every experiment autoencoder was used to generate representations for the input data on top of which a logistic regression model was trained for building a classification model. Some of the results on CIFAR-10 dataset are presented in plots from Figure 2.

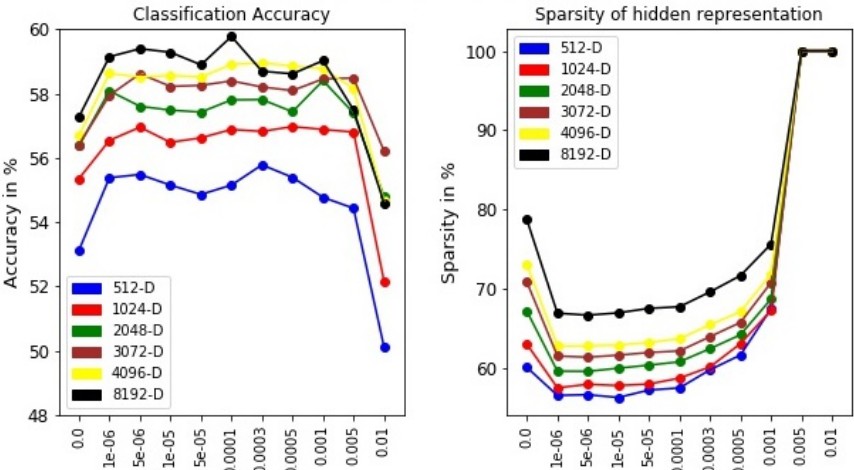

Figure 3: Sparsity trend in hidden representations of single-hidden-layer MLP trained on PCA reduced CIFAR-10 dataset and classification accuracy w.r.t regularization strength of OVR-loss. Legend in each plot states the number of hidden units.

From figure 2, it can be observed that increase in $\lambda$ (regularization strength for OVR-loss) till a certain level increased both sparsity in representations and its classification accuracy. Increase in $\lambda$ beyond $10^{-4}$ still increases the sparsity in representations but with decrease in classification accuracy.

We also ran supervised classification experiments on CIFAR-10 dataset using single hidden layer MLP models with OVR-loss as regularization and the results are presented in Figure 3. From the plots it can be observed that sparsity of the hidden layer representation increases with increased regularization strength and also using OVR-loss as regularization we achieve performance on-par with network using Dropout Hinton et al. (2012). Please refer to appendix for

Table 1: Classification performance of Logistic regression model on representations learned using different models on PCA reduced CIFAR-10 dataset

| MODEL | ACCURACY |
|---|---|
| DAE | 57.79 |
| OVR-Encoder | 57.65 |
| K-means | 49.46 |
| Only Logistic | 39.78 |

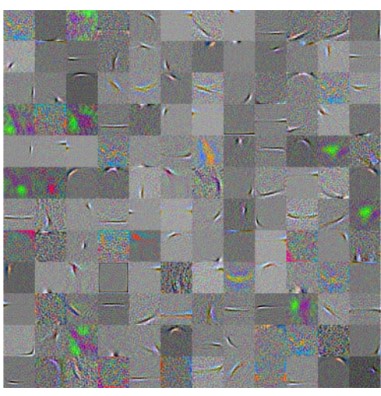

Table 2: Weight features learned by OVR-encoder with 8192 neurons and trained on PCA reduced CIFAR-10 dataset.

more details on the experiments along with experiments involving denoising-autoencoders and MLPs with Dropout, L1 and L2 activity regularizers.

## 4 A SINGLE-LAYERED ENCODER MODEL

Given that efficient representation of data can be achieved by simply making sure that different regions or clusters of a discontinuous input manifold are being mapped to representations with least overlap, we propose a new unsupervised single-layer network, "OVR-Encoder", with cost function as OVR-loss (from Equation 1). When we use OVR-loss as cost function, one likely trivial solution is that all hidden activations are pushed to zero which reduces the error to zero. To prevent this, we add another term $L$ to the cost function for encouraging non-zero activations in the network.

$$L = |\frac{\sum H}{n} - 0.5| \tag{2}$$

So the final cost function $J$ is a weighted sum of OVR-loss and $L$ from Equation2,

$$J = |\frac{\sum H}{n} - 0.5| + \lambda \sum HH^T \tag{3}$$

OVR-loss based update rule for the weight vector $\vec{w}_k$ which corresponds to the $k^{th}$ neuron in the hidden layer is,

$$\Delta\vec{w}_k \sim -\sum_{j\in N} h_k^j \sum_{i\in N, i\neq j} \vec{x}^i \tag{4}$$

where $N$ is the batch of input samples $\vec{x}$, $h_k^j$ is the response of neuron $k$ for input $\vec{x}_j$ from the batch. Interestingly, while single-layer models like online-k-means MacQueen et al. (1967) update their weights/centroids in the direction of input, our model pushes the weights, as shown in Equation 4, away from the weighted mean of rest of the samples from the batch. We intend to study the effect of this nature of our model more in the future work.

To evaluate the effectiveness of the learned data representations from the OVR-Encoder, we use the same dataset and experimental pipeline as described in Section 3. We trained OVR-Encoder with 8192 neurons for learning representations of PCA reduced CIFAR-10 data. During training, batch size of 128 and Adam optimizer with an initial learning rate of 0.001 was used. The regularization strength $\lambda$ was varied between $10^{-5}$ to $5 \times 10^{-4}$. The performance of the Logistic Regression model on representations from the best OVR-Encoder is presented in Table 1 along with performances of Logistic regression model trained on representations from different models. The OVR-Encoder model performs poorly when the training batch size is too small as the OVR-loss operates by computing overlap across the representations of multiple input samples. We also visualize the learned weights or features vectors from the OVR-Encoder model in figure from Table 2. It can observed that the features learned are mostly local Gabor filters, very similar to ones learned in regularized autoencoder models.

## 5 CONCLUSION

Many approaches have been proposed for learning sparse representations and its importance in neural networks. We in this article presented a reasoning for emergence of sparsity and its relation to the input data distribution, "more discontinuous the input manifold more sparse the representations" (see Section 2). By employing OVR-loss (see Section 3) for regularization in autoencoders and MLPs, we achieved encouraging results which support our hypothesis on sparsity. We showed that efficient representations of data can be learned simply by minimizing OVR-loss only and proposed a single layer encoder only model (OVR-Encoder) which uses Hebbian-like local learning rules for training, and does not require error-backpropagation. In future, we intend to understand the novel learning approach of the OVR-Encoder and explore its usability for continuous learning problems. Also, understanding sparsity in multi-layer models and using OVR-loss for training multi-layer networks will be part of our future steps.

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

## 6 APPENDIX

### 6.1 AUTOENCODER EXPERIMENTS

To compare OVR-loss as regularization against other form of regularization we trained single hidden layer DAE and OVR auto-encoders with multiple hidden layer size and regularisation strength configurations on PCA reduced CIFAR-10 data with 50000 training sample and 10000 validation samples. The number of hidden units were varied between 2048 and 8192 for both DAE and OVR-AE. For OVR-AE, the regularization strength $\lambda$ was tried with values between $10^{-6}$ to $10^{-2}$. For DAE, the dropout ratio for input layer was varied with $0.2, 0.3, 0.4, 0.5$ values. For OVR-AE, sigmoid and relu activations were used in the hidden layer, and for DAE, sigmoid activation was used. Each configuration was trained with Adam optimizer with an initial learning rate of $0.001$ with a scheduler to reduce the learning rate when the training loss hits a plateau. The model at the epoch with least validation loss was saved for computing the hidden layer representations of the data. For stability of the training process for computing the OVR-loss we used hidden representations normalized to unit vectors.

We trained a logistic regression model on the hidden layer representations from each autoencoder trained above and their classification accuracy are presented in Table4.

### 6.2 MLP EXPERIMENTS

To show that sparsity in the hidden layer representations of an MLP network trained with regularization is proportional to the regularization strength, we ran a series of experiments on same dataset described in previous section. We used various regularization techniques (dropout, OVR, L1 and L2 Activity regularizers) for the experiments and also observed the corresponding classification performance. The number of hidden units were varied between $512$ and $8192$ for all configurations. For OVR, the regularization strength $\lambda$ was varied between $10^{-6}$ to $10^{-2}$. For dropout, the input dropout ratio was varied with $0.0, 0.2, 0.3, 0.4, 0.5$ values and the hidden dropout ratio was fixed at $0.5$. For L1 and L2 Activity regularizers, the regularization strength was varied between $10^{-6}$ to $10^{-2}$. For activation function, sigmoid and relu were used. Each network configuration was trained with an input dropout ratio of $0.2$ (for OVR, L1 and L2) and Adam optimizer was used with an initial learning rate of $0.001$. All the networks were trained for $75$ epochs with a scheduler to reduce the learning rate when the training loss hits a plateau. In the case of CIFAR-10, we use PCA based dimensionality reduction without whitening for preprocessing.

The sparsity plots corresponding to the dropout validation experiments are shown in Figure 5. From Figures 4,5 it can be observed that the proportionality between the input noise level and the corresponding hidden layer sparsity holds in the dropout experiments as well as DAE experiments. In Konda et al. (2015a) the authors show that dropout noise scheme can also be seen as data augmentation approach which can result in increased number of input regions to be represented, thereby increased sparsity of the hidden layer representations. In the case of Figure 5 (a) the sparsity initially increases with the input noise level but then starts to fall. This may be due to the dense PCA representation of the input data, where higher noise levels probably destroy the input manifold structure.

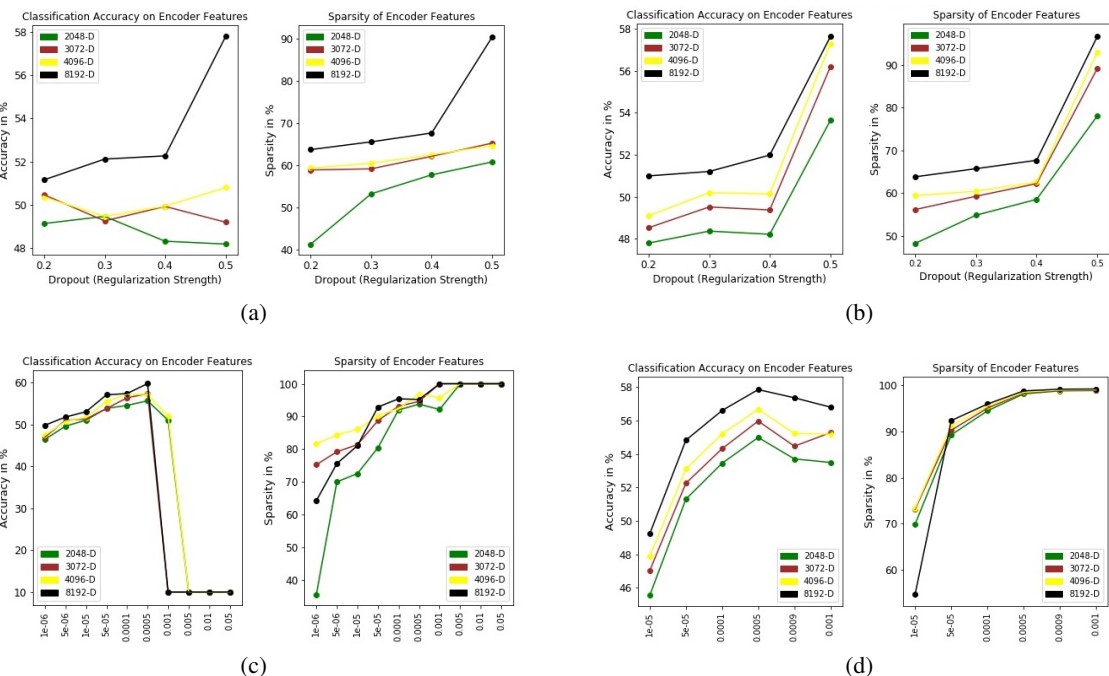

Figure 4: Sparsity and classification accuracy w.r.t regularization strength in autoencoder experiments, using ReLU and sigmoid activation with OVR-loss as regularization, trained on PCA reduced CIFAR-10.. **Row 1** DAE and **Row 2** OVR. Legend in each plot shows InputDimensions-hiddenUnits in the experiment to the curve.

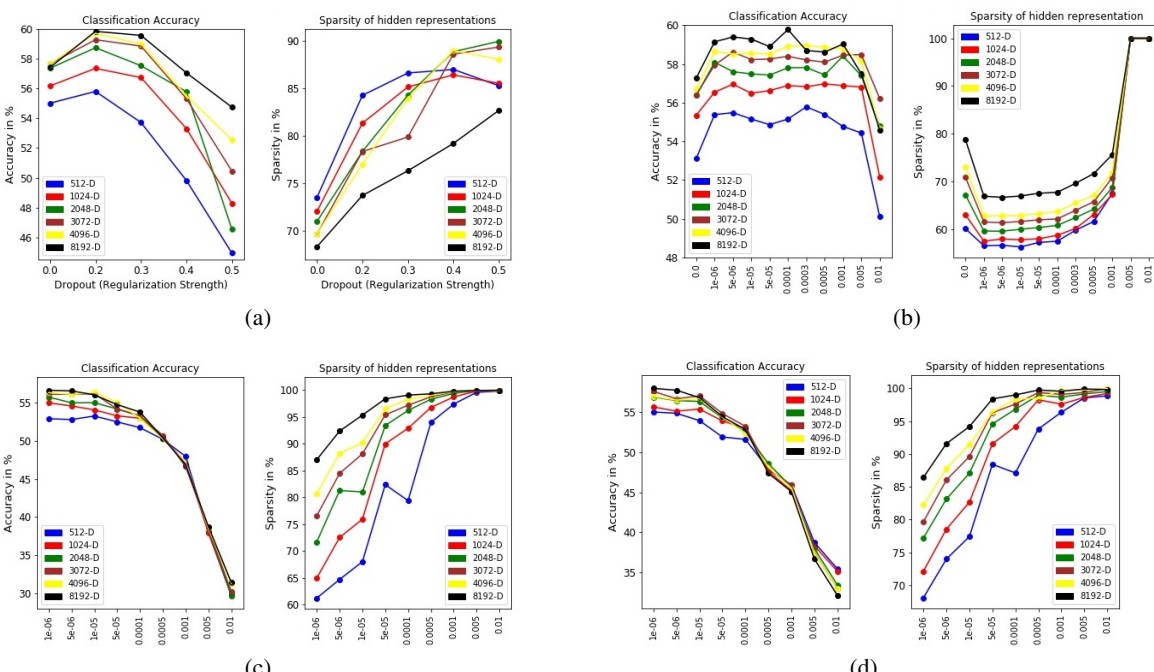

Figure 5: Sparsity trend in hidden representations and classification accuracy w.r.t regularization strength of single-hidden-layer MLP trained on PCA reduced CIFAR-10 dataset. (a) Dropout (b) OVR-loss (c) L1-norm (d) L2-norm. Legend in each plot tells the hidden layer size.

