# OpenReview forum: "ONLY SPARSITY  BASED  LOSS FUNCTION FOR  LEARNING  REPRESENTATIONS"
_ICLR.cc/2019/Workshop/LLD — Submitted to LLD 2019_

### Official Review · AnonReviewer2 · 2019-04-08
**The paper proposes an interesting loss ruction that relies on independence of datapoints, unfortunately, it also depends on the data to work.**

**Rating:** 3
**Confidence:** 2

**Review:**

The authors propose a loss function that encourages different samples to be mapped to different feature vectors. The loss is analyzed both in the context of an autoencoder and a single layer supervised neural network.
One issue the authors would consider is introducing more clarity in the exposition of the work. For instance, you never really specify the exact inputs and outputs of your autoencoder.
My main criticism would be that OVR overly relies on the quality of the neighboring samples in a minibatch to identify the loss. if you have a mini-batch of datapoints of the same class it would not work at all. However, even in the case where you have a minibatch of 128 and an expected 12 data points of each class for CIFAR-10 I would expect a lot of "noisy information" steering the loss towards the wrong result. That problem might alleviate in the  case of many labels and highly diverse datasets.
I think in total the work leaves a lot to be desired. 1. The loss can be broken by the right construction of the dataset. 2. The loss might do better with a rich dataset which is probably not the case for most applications discussed in the LLD workshop.

---

### Official Review · AnonReviewer1 · 2019-04-13
**Review of "ONLY SPARSITY BASED LOSS FUNCTION FOR LEARNING REPRESENTATIONS"**

**Rating:** 1
**Confidence:** 2

**Review:**

The paper discusses a regularisation function that encourages sparsity. Then, the authors use this criterion as a loss to train neural networks in the unsupervised setting. Numerical experiments comparing the approach with other regularization functions are provided.

First of all, it seems there are strong formatting issues with the paper. The conclusion is on the 5th page, and the margins are significantly larger than other submitted paper.

Globally, all the choices of the paper are poorly motivated, as well as the assumptions. It is hard for the reader to be convinced by the (many) claims in this paper. On top of it, the experiments are not convincing at all, as the dropout seems to be as efficient as the proposed approach (in terms of the tradeoff between accuracy and sparsity).

I have many concerns about the author's motivations for the chosen loss. As far as I understand, they penalize the matrix H using a quadratic function (sum of elements in HH^T). However, this seems to me not straightforward why it should encourage sparsity, since
1) the diagonal elements are taken into account in the regularisation and
2) the function is quadratic.

Finally, the writing of the paper seems to have been done in haste, as the explanations are not clear at all, and the math formulas are incomplete.

---

### Decision · Program_Chairs · 2019-04-16
**Acceptance Decision**

Reject